# Human iPSC-Derived 2D and 3D Platforms for Rapidly Assessing Developmental, Functional, and Terminal Toxicities in Neural Cells

**DOI:** 10.3390/ijms22041908

**Published:** 2021-02-14

**Authors:** Ileana Slavin, Steven Dea, Priyanka Arunkumar, Neha Sodhi, Sandro Montefusco, Jair Siqueira-Neto, Janet Seelke, Mary Anne Lofstrom, Blake Anson, Fabian Zanella, Cassiano Carromeu

**Affiliations:** 1StemoniX, La Jolla, CA 92037, USA; Ileana.Slavin@stemonix.com (I.S.); steven.dea@stemonix.com (S.D.); priyanka.arunkumar@stemonix.com (P.A.); neha.sodhi@stemonix.com (N.S.); janet.seelke@stemonix.com (J.S.); maryanne.lofstrom@stemonix.com (M.A.L.); blake.anson@stemonix.com (B.A.); fabian.zanella@stemonix.com (F.Z.); 2Center for Discovery and Innovation in Parasitic Diseases, Skaggs School of Pharmacy and Pharmaceutical Sciences, University of California San Diego, La Jolla, CA 92093, USA; smontefusco@health.ucsd.edu (S.M.); jlagedesiqueiraneto@health.ucsd.edu (J.S.-N.)

**Keywords:** human induced pluripotent stem cells (hiPSCs), drug discovery, neurodevelopmental toxicity, repurposed drugs, organoids

## Abstract

With increasing global health threats has come an urgent need to rapidly develop and deploy safe and effective therapies. A common practice to fast track clinical adoption of compounds for new indications is to repurpose already approved therapeutics; however, many compounds considered safe to a specific application or population may elicit undesirable side effects when the dosage, usage directives, and/or clinical context are changed. For example, progenitor and developing cells may have different susceptibilities than mature dormant cells, which may yet be different than mature active cells. Thus, in vitro test systems should reflect the cellular context of the native cell: developing, nascent, or functionally active. To that end, we have developed high-throughput, two- and three-dimensional human induced pluripotent stem cell (hiPSC)-derived neural screening platforms that reflect different neurodevelopmental stages. As a proof of concept, we implemented this in vitro human system to swiftly identify the potential neurotoxicity profiles of 29 therapeutic compounds that could be repurposed as anti-virals. Interestingly, many compounds displayed high toxicity on early-stage neural tissues but not on later stages. Compounds with the safest overall viability profiles were further evaluated for functional assessment in a high-throughput calcium flux assay. Of the 29 drugs tested, only four did not modulate or have other potentially toxic effects on the developing or mature neurospheroids across all the tested dosages. These results highlight the importance of employing human neural cultures at different stages of development to fully understand the neurotoxicity profile of potential therapeutics across normal ontogeny.

## 1. Introduction

With the rise in incidence of global health threats, screening systems able to accelerate the discovery of effective and safe new treatments are vitally important. Central nervous system (CNS) toxicity is a major concern in drug development and a frequent cause of attrition in late-stage clinical trials as well as in the withdrawal of already approved drugs from the market [1]. The current drug development pipeline typically involves compound testing in immortalized cell lines followed by in vivo animal investigations; results are then extrapolated to humans. While convenient and providing high-throughput access, the biology of immortalized cultures can be altered by the transformation process and continued long-term culture [2]. Furthermore, because immortalized cell lines are pure cultures, they do not reflect the biological complexity of the tissue. Animal models have proved valuable for identifying disease mechanisms and genetic function; however, they are ill-suited for testing large numbers of compounds, and recent studies have magnified the differences in both molecular and cellular neural composition amongst species [3,4], which may underlie the poor track record of translation to humans.

To move into a more relevant environment, scientists are using human-induced pluripotent stem cell (iPSC)-derived tissues [5,6,7]. These cellular systems offer a human-based environment while maintaining a practical balance between scalability and biological complexity [8]. We have built on these advances by creating 2D and 3D high-throughput human iPSC-derived neural platforms that have demonstrated utility in discovery- and toxicity-based screening applications [9,10]. Because the developing fetal nervous system is more vulnerable to chemical and biological assault than the mature system, it is vital to understand the potential for neurodevelopmental toxicity from compounds deemed safe through testing on mature cells. To that end, we modified our platforms to enable potential neurotoxicity assessment at different stages of human neurodevelopment from the progenitor cell stage to a mature, functional neural network.

There are no current, standardized protocols for testing developmental neurotoxicants [11]; thus we expanded our previous neurotoxicity study in mature neurons and astrocytes [9] by investigating the potential impact of compounds on neurodevelopment in attached monolayer (2D) and suspension spheroid (3D) hiPSC-derived neural models that mimic aspects of in vivo CNS development. As an initial proof-of-concept, 29 compounds, previously identified as potential antiviral therapeutic candidates for treating Zika Virus (ZIKV) infection and potentially preventing the associated devastating neurodevelopmental consequences, were investigated for their toxic impact on both viability and function of developing and mature neural cells. Our data showed that progenitor cells were far more susceptible to compound toxicity than their mature counterparts, that functional toxicity can manifest at lower concentrations than decreased cell viability, and that 2D and 3D assays can be used in complementary experiments to answer different questions about the physiology of the developing human CNS. Together, these data demonstrate the importance of using HTS platforms able to quickly assess compound safety at different stages of human neurodevelopment in vitro.

## 2. Results

### 2.1. iPSC Neural Cultures Model Neurodevelopment in High-Throughput

Two- and three-dimension cell culture systems that recapitulate different stages of neural development (progenitor and mature) were adapted to screen for neurodevelopmental toxicity (NDT) across two independent iPSC clones. The overall experimental approach (Figure 1A) was to differentiate neural progenitor cells (NPCs) to form 2D monolayer or 3D neurospheroid cultures of neurons and astrocytes. NPC identity was confirmed through immunocytochemical staining for Nestin and SOX1 (Figure 1B) while astrocytic and neuronal cell populations were confirmed with GFAP and MAP2 staining, respectively, in both 2D and 3D formats (Figure 1C,D). Additional characterization demonstrated NPC clonal equivalency through flow cytometry showing 95.3 and 98.3% NPC purity (Appendix A) as well as neuronal maturity through immunocytochemical labeling of mature synapses with anti-synapsin antibodies in both 2D and 3D formats (Appendix A) and neuronal nuclei with anti-NeuN antibodies (Appendix A).

### 2.2. Compound Incubation Duration Affects NPC Viability

As proof of concept to illustrate 2D and 3D neural cultures as a testing platform for detecting NDT, we selected 29 compounds previously identified in the literature as potential therapeutics to treat ZIKV infection (Table 1).

Compound exposure time was optimized by exposing neural progenitor cells to compounds for 24, 48, and 72 h followed by viability testing using CellTiter-Glo reagent. While many of the detrimental effects were evident after 24 h of exposure, for some compounds, a more pronounced effect was only apparent after 72 h of incubation (Appendix A). For example, exposure to Dinaciclib, a cyclin-dependent kinase (CDK) inhibitor, showed viability reductions to 78 ± 2%, 34 ± 3%, and 15 ± 5% of control levels after 24, 48, and 72 h of incubation at 10 µM, respectively (mean ± s.d. and *n* = 3 each condition), and Dactinomycin, a chemotherapeutic agent, showed viability reductions to 71 ± 4%, 31 ± 3%, and 17 ± 7% of control levels after 24, 48 and 72 h of incubation at 10 µM, respectively (mean ± s.d. and *n* = 3 for each condition). Conversely, exposure to Gemcitabine, a chemotherapeutic agent, led to much greater cell death at 24 h with less pronounced increases at 48 and 72 h of incubation at 10 µM with viability reductions to 25 ± 4%, 8 ± 4%, and 20 ± 23% of control levels, respectively (mean ± s.d. and *n* = 3 for each condition). Similar data were obtained from an independent NPC clone after 72 h of compound incubation using PrestoBlue as the cell viability indicator (Appendix A, left panel). These data suggest that compound exposure time as well as compound concentration influences toxicity. To ensure the most notable effects were captured, all subsequent viability measurements were performed after 72 h of compound incubation. Mean ± s.d. values for all conditions can be found in Appendix A.

### 2.3. Neural Maturity Influences Sensitivity to Compound Toxicity

The influence of developmental stage on compound-induced toxicity was determined by profiling compound effects at the NPC stage as well as at 2, 4, and 8 weeks of differentiation in 2D neural cultures (Figure 2). Following maturation to the prescribed age, cells were exposed to compounds for 72 h and then assayed for viability. A progressive decrease in sensitivity to the applied compounds was observed as the cells differentiated from the progenitor stage (NPC) through 8 additional weeks of development. Several compounds, including dactinomycin, dinaciclib, RGB-286147, and bortezomib, were observed to severely compromise NPC viability but showed a remarkably less drastic impact on the viability of 8-week old neural cultures with NPCs showing viability levels of less than 35% and 8 week old cultures showing viability levels greater than 80% for these compounds at 10 µM. ANOVA analyses of the mean response at 0.01 µM and 10 µM showed significant differences across the tested ages at both concentrations (*p* = 0.017 and 7.9 × 10^−9^, respectively). Mean ± s.d. values and statistical tests can be found in Appendix A. Together, these results suggest that the cellular developmental stage is an important contributor to toxicity susceptibility.

### 2.4. Culture Format Does Not Have a Significant Impact on Compound-Induced Cell Death

Three-dimension cell culture formats are gaining popularity as they provide an in vitro format that is more comparable to native tissue [14]. To determine the influence of cell culture format on compound-induced cell death, 2D neural cultures and 3D neurospheres at both immature and mature culture ages were interrogated with the 29 compounds followed by CellTiter-Glo (CTG)-based viability assessment. As shown in Figure 3, immature neural cultures are more susceptible than their mature counterparts to compound-induced toxicity, but culture format had virtually no impact on compound susceptibility. T-tests comparing similar aged cultures across 2D or 3D formats showed no significant differences at the analyzed mid (0.01 µM) and high (10 µM) dose levels (*p* > 0.05). Mean ± s.d. values and statistical tests can be found in Appendix A. Using brightfield microscopy (Appendix A), we also observed severe morphological changes to both 2D and 3D cultures following treatment with 10 µM compound for 48 h. As CTG measures ATP content and is a surrogate marker for cell death, we confirmed cell death through caspase 3/7 activation (Figure 4 and Appendix A).

### 2.5. 2D and 3D Mature Neural Models Detect Functional Toxicity

Viability assays are typically terminal readouts that measure the end-stage toxicity phenotype, i.e., cell death [15]. Functional toxicity can happen at lower doses and earlier time points [9]. Therefore, the functional impact of compounds on mature 2D and 3D neural cultures was assessed using a calcium mobilization assay performed in a high-throughput Fluorometric Imaging Plate Reader (FLIPR) as a surrogate biomarker for neuronal and synaptic activity [9,10]. A selection of toxic and non-toxic compounds, based on their previously determined impact on viability, were chosen for testing on 8-week old 2D and 3D neural cultures with the 3D cultures receiving a range of compound concentrations and the 2D cultures receiving the no observed adverse effect level (NOAEL) as determined in the viability assays (Table 2).

Two-dimensional neural cultures have previously been shown to have synchronized activity [10]; however, such activity can be less regular than that of 3D cultures. Stimulation with an agonist, such as glutamate (Figure 5A) or kainate (Appendix A) elicits a strong synchronized burst of activity across the culture. When applied in the presence of test compound, changes in the glutamate- or kainate-induced synchronized burst offer an indication of the functional impact, or toxicity, of the test compound. Treatment of 2D cultures with three of the 14 compounds (ivermectin, mefloquine, and sertraline) significantly impaired the response of the 8-week old neural cultures relative to controls in both glutamate and kainate challenges (Figure 5B and Appendix A, respectively; *p* < 0.0001 for all three compounds, ANOVA followed by Dunnett’s multiple comparison test) suggesting that, while these three compounds may not adversely impact cell viability during development at their NOAEL dosage, there could be undesirable functional changes in neuronal activity.

Three-dimension neurospheroids have been previously shown to exhibit spontaneous, highly synchronized, neural activity that can be measured as an oscillating Ca^2+^ waveform in the presence of a Ca^2+^ fluorophore without the need for a synchronizing excitatory challenge [9,10]. Using the same compounds as in the 2D investigations, the impact on neural activity was determined before and after compound addition (Figure 6). Interestingly, differences in the impact of compounds were evident on the 3D (spontaneous activity) versus the 2D (induced activity) cultures. For example, 3D cultures were more sensitive than 2D cultures to ivermectin, palonosetron, and pyrimethamine; neither palonosetron nor pyrimethamine showed any impact on 2D cultures, but both showed significant functional decrease in 3D culture activity (*p* < 0.0001 ANOVA followed by Dunnett’s multiple comparison test). Conversely, 2D cultures were more sensitive than 3D culture to mefloquine and sertraline; in this case, neither compound showed an effect on 3D cultures. These results indicate the need for a broad investigative approach, using complementary assays (induced and spontaneous activity) when interrogating modulation of the CNS. Mean values and statistical tests can be found in Appendix A.

## 3. Discussion

While clinical compounds have typically been shown to be effective in established cultures and adult animals, the possible NDT effects of these compounds remain largely unexplored [16]. The translation of neurodevelopmental toxicity studies from rodents to humans is particularly challenging, especially at later stages of embryonic development and placentation [17,18]. Gene expression studies comparing rodent and human placentae highlight the divergence at later stages of pregnancy [19]. Thus, there is a clear need for alternative human-based neurodevelopmental models to improve predictors for adverse outcomes. The creation and availability of hiPSCs and hiPSC-derived neural cultures have opened new possibilities for studying drug-induced neurotoxicity, including adverse effects occurring in early neural development [14].

Here, we developed physiologically-relevant human neural culture systems to assess multiple neurodevelopment stages (progenitor, 2 weeks, 4 weeks, and 8 weeks of differentiation) in different formats (2D neural cultures and 3D neurospheroids) and, using a combination of viability and functional assays, quickly and economically evaluated possible NDT effects of 29 potential anti-viral compounds in a high-throughput fashion. Viability assays performed using human iPSC-derived cells over the range of neural differentiation stages indicated that early neural cultures at the progenitor stage showed greater susceptibility to compound-mediated changes in cellular viability than more mature neural cultures irrespective of 2D or 3D culture format.

The greater susceptibility of NPCs to compound-induced decreased viability suggests that additional preclinical testing should take into account developing organ systems for target patient populations such as expectant mothers. Additionally, the increased sensitivity of progenitor cells may have an impact on mature systems. Neural Stem Cells, (NSCs) are the in vivo equivalent of NPCs found in the adult brain where they are thought to contribute to injury repair [20,21], and this population shares a similar sensitivity to toxins as NPCs [22]. Thus compounds with an NDT liability but deemed safe for use in mature contexts, may in fact have effects during injury repair. Thus the principles of NDT described here may need to be considered across broader therapeutic contexts.

As viability studies examine late-stage toxicity, 13 of the 29 compounds were chosen for further functional analysis at their NOAEL for viability to look for underlying functional toxicity (Figure 5 and Figure 6). Two-dimension neural culture activity was synchronized with a neuroexcitatory challenge (glutamate or kainate) while 3D neurospheroid activity was spontaneously synchronized and did not require any additional triggers [10]. Not surprisingly, both the 2D and 3D platforms detected functional toxicity at the NOAEL for viability. Unexpectedly, only one compound modulated both systems. Ivermectin, mefloquine, and sertraline all had significant functional impacts when assayed in the 2D-induced platform while cyclosporin A, ivermectin, palonosetron, and pyrimethamine had significant functional impacts in the 3D-spontaneous platform. Notably, ivermectin, which had impact in both systems, has been linked to broad neuronal interactions across GABA receptors, nicotinic acetylcholine receptors, glycine receptors, and chloride channels [23,24].

Mefloquine and sertraline only modulated induced activity in the 2D system. Mefloquine is labeled by the FDA as having neurologic side effects [25], while sertraline may have its observed effect in induced response through its mechanism of action, acting by inhibiting the uptake of serotonin by the neural network [26]. Cyclosporine A, palonosetron, and pyrimethamine all only modulated spontaneous activity in the 3D system. Clinically, cyclosporine A is known to induce neurological side effects in up to 40% of patients [27]. Similar to sertraline, palonosetron acts on the serotonergic system but as a 5-hydroxytryptamine 3 receptor antagonist [28]. Finally, pyrimethamine has been associated, with headache (10% or more of patients), dizziness (1% to 10% of patients), and seizures (less than 0.01% of patients) [29]. While potential CNS links can be established for all of the compounds showing functional impact, it is less clear why some compounds only show effects on a single platform. The selective impact may be due to a compound’s specific pharmacology and mechanism of action relating to the underlying functional context of the assay, e.g., being detectable during massive depolarization (the 2D platform) or spontaneous oscillatory behavior (the 3D platform).

The present study does not incorporate a blood-brain barrier, thus the actual compound concentration seen by the in vitro neural preparations may be higher than that observed in vivo. While the blood-brain barrier is crucial in protecting neural tissue from exogenous compounds, it is not expected to impact neural response for a given concentration that comes into contact with the neural tissue. Thus, the primary conclusions of this study, namely that progenitor cells are more susceptible to toxic insult, that functional responses are more sensitive readouts of toxic insult than terminal viability endpoints, and that different functional assays detect different compound responses, still stand, can be understood in the absence of a blood-brain barrier, and are worthy of consideration during early development. Together, our results demonstrate the complementary nature of the assays and highlight the potential need for multiple investigations to accurately identify compound liabilities.

## 4. Materials and Methods

### 4.1. Human iPSC-derived Neural Cultures

StemoniX produced and commercially supplied microBrain 3D Assay Ready 384-Plates and microBrain 2D Assay Ready 384-Plates, pre-plated human cortical neural co-cultures used in this study (StemoniX, Maple Grove, MN, USA). Each well of the microBrain 2D platform contains a balanced mixture of iPSC-derived cortical neurons and astrocytes matured from StemoniX NPCs. Each well of the microBrain 3D platform plate contains a single, free-floating 3D neurospheroid of similar composition [9,10].

### 4.2. Neural Differentiation and Maintenance

Progenitor cells were kept in Neural Basal Media (DMEM/F12 basal media supplemented with 0.5X N2 supplement, 0.5X B27 supplement and 1X Penicillin-Streptomycin (all reagents from Thermo Fisher Scientific, Waltham, MA, USA) supplemented with 20 ng/mL of FGF-2 (Thermo Fisher Scientific, Waltham, MA, USA). Neural cultures were maintained in BrainPhys Media (StemCell Technologies, Vancouver, BC, Canada) supplemented with 20 ng/mL of BDNF (R&D Systems, Minneapolis, MN, USA) and 20 ng/mL of GDNF (R&D Systems, Minneapolis, MN, USA) and allowed to differentiate for 8 weeks. Every 2–3 days, 50% partial media changes were performed.

### 4.3. Compound Administration

Master stock solutions of all compounds were prepared by resuspending and diluting compounds to 10 μM stock concentrations following manufacturers’ instructions (Appendix A). Prior to cell treatment, compounds were diluted in either Neural Basal media (for NPCs) or BrainPhys media (for neurons) to achieve the final desired concentrations.

### 4.4. Cell Viability

Cell viability was evaluated using the CellTiter-Glo cell viability kit (Promega, Madison, WI, USA) and the PrestoBlue cell viability kit (Thermo Fisher Scientific, Waltham, MA, USA) according to the manufacturers’ directions. Cell viability performed on human iPSC-derived NPCs and neural cultures was analyzed 24, 48, and 72 h after compound addition using the SpectraMax i3 (Molecular Devices, San Jose, CA, USA) and following the manufacturer’s instructions.

### 4.5. No Observed Adverse Effect Level (NOAEL) Concentration Determination

To assess the functional toxicity of compounds, we selected compounds with diverse cell viability impact (highly toxic and non-toxic) and determined their NOAEL concentrations. The data used to generate these concentrations were from 2D cultures of NPCs. For each compound, we calculated the lowest concentration of compound able to provide 85% or greater viability (Appendix A).

### 4.6. Caspase-3/7 Activation Assay

Caspase-3/7 activation was measured after 48 h of incubation with compounds using the CellEvent^TM^ Caspase-3/7 Green Detection reagent following the manufacturer’s instructions (Thermo Fisher Scientific, Waltham, MA, USA). In brief, cells were incubated with compounds for 48 h before the assay. On the day of the assay, the Caspase-3/7 reagent was mixed with complete BrainPhys media to a final concentration of 5 μM and added to cells. After incubation for 30 min at 37 °C, cells were imaged using an ImageXpress Confocal microscope (Molecular Devices, San Jose, CA, USA).

### 4.7. Immunostaining

Human iPSC-derived NPCs and neural cultures were fixed using 4% paraformaldehyde for 10 min and washed three times with PBS 1X without Ca^2+^ and Mg^2+^. Cells were permeabilized with PBS^+^ with Ca^2+^ and Mg^2+^/0.2% Triton X-100 for 15 min at room temperature and subsequently blocked using Odyssey Blocking Buffer (LI-COR, Lincoln, NE, USA) for 4 h at room temperature. Cells were then incubated overnight with primary antibodies (Table 3). Following incubation, cells were washed three times with PBS^+^ with Ca^2+^ and Mg^2+^/0.1% Tween20 and incubated for 1 h with Alexa Fluor conjugated secondary antibodies (Thermo Fisher Scientific, Waltham, MA, USA) at room temperature. Cells were then washed once with PBS^+^ with Ca^2+^ and Mg^2+^/0.1% Tween-20 and incubated with Hoechst for 4 min at room temperature for nuclei staining. Finally, cells were washed three times with PBS^+^ with Ca^2+^ and Mg^2+^/0.1% Tween-20 and left at 4 °C until imaged.

### 4.8. Imaging

Immunostained NPCs and neural cultures were imaged using an ImageXpress Confocal microscope (Molecular Devices, San Jose, CA, USA). Images were edited using the ImageJ software imaging collection package FIJI.

### 4.9. Calcium Mobilization Assays

Calcium oscillation assays were performed as described in our previous publication [9]. Prior to compound addition, cells were loaded with Calcium 6 dye (Molecular Devices, San Jose, CA, USA) according to the manufacturer’s specifications. To assess the impact on neural functionality, compounds were added and incubated for 30 min before the assay. All recordings (2D and 3D) were done in a FLIPR^TETRA^ system (Molecular Devices, San Jose, CA, USA). For induced activity (2D), cells were challenged with glutamate or kainate, both at 100 μM. For 3D cultures, neural spontaneous activity was recorded for 10 min. Data were analyzed using Peak Pro software (Molecular Devices, San Jose, CA, USA).

### 4.10. Graphs and Statistical Analysis

Comparisons between two populations were performed using the Student’s t-Test (Figure 3), comparisons across a group of populations (>2) were performed using one-way analysis of variance (ANOVA) (Figure 2, Figure 5B and Figure 6, Appendix A), and finally comparisons between individual populations within a group (> 2) were performed using Dunnett’s multiple comparison test following a one-way ANOVA. The mean, standard deviation, and number of replicates for each heat map graph and its associated statistical comparison, as well the multiple comparison statistical results from the FLIPR data are detailed in Appendix A. *p* values < 0.05 were considered significant. All graphs and statistical analyses were done using GraphPad Prism version 9 software (GraphPad, San Diego, CA, USA).

## 5. Conclusions

To identify potential neurodevelopmental toxicity of compounds, early progenitor and differentiated human iPSC-derived neural cultures were used to analyze the safety profiles of compounds previously reported to have satisfactory anti-ZIKV activity. The data presented here indicate that of the 29 drugs tested, only 4 (daptomycin, emricasan, methoxsalen, and roscovitine) displayed acceptable neurotoxicity profiles that did not affect neural viability and functionality at any developmental stage in all the concentrations tested. Altogether, our results highlight the necessity of employing human neural culture models at different stages of maturation to assess both cell viability and function to test the neurodevelopmental toxicity of drugs under evaluation for potential therapeutic use.

## Figures and Tables

**Figure 1 ijms-22-01908-f001:**
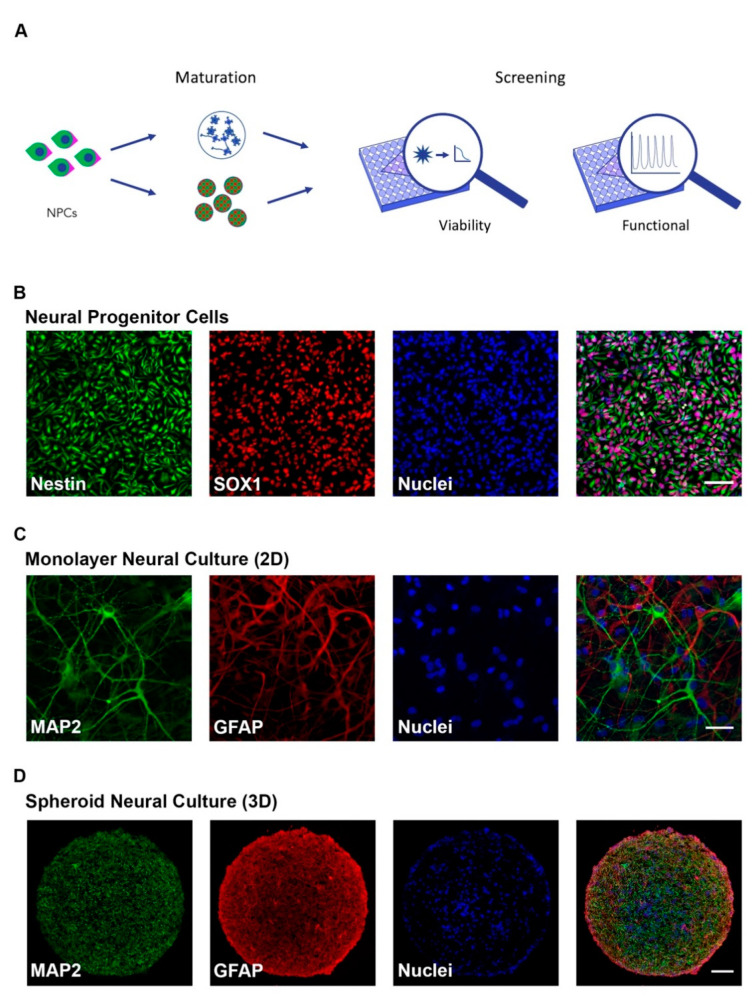
Characterization of human induced pluripotent stem cell (hiPSC)-derived neural cultures. (**A**) Schematic of experimental design. Human iPSC-derived neural progenitor cells (NPCs) were matured in 2D and 3D formats and cellular viability (CellTiter-Glo or PrestoBlue) and functional response (FLIPR^TETRA^) screenings to compounds were performed; (**B**) Immunocytochemistry (ICC) of neural progenitor cells showing the presence of Sox1 and Nestin (Scale bar = 50 μm); (**C**,**D**) ICC of 8-week-old 2D neural cultures (**C**) (Scale bar = 50 μm) and 8-week-old 3D neural spheroids (**D**) (Scale bar = 100 μm) showing the presence of maturity markers for neurons (MAP2) and astrocytes (GFAP).

**Figure 2 ijms-22-01908-f002:**
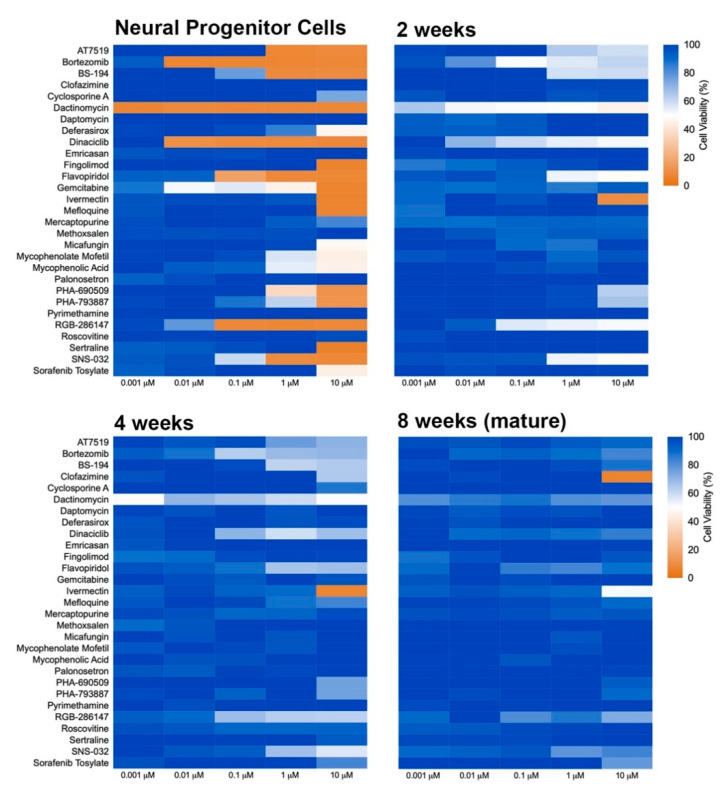
Heat map of toxicity profile over the time of neural differentiation. NPCs were plated in 384-well plates and matured for 2, 4, and 8 weeks. Neural cultures at different ages of maturation were exposed to compounds for 72 h, and viability was determined using PrestoBlue. Graphs show heat map for cell viability compared to control (DMSO-treated).

**Figure 3 ijms-22-01908-f003:**
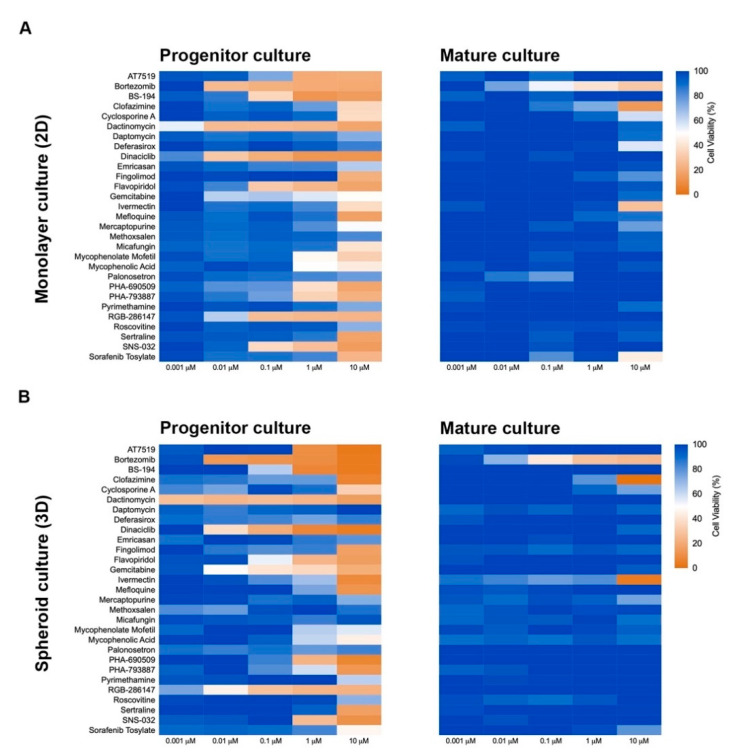
Culture format (2D or 3D) does not influence cellular toxicity response to compounds. Heat maps of cellular viability compared to control (DMSO-treated) after exposure to compounds for 72 h. Immature (NPC) and mature (8-week-old) 2D (**A**) and 3D (**B**) neural cultures are shown. Cellular viability was determined using CellTiter-Glo.

**Figure 4 ijms-22-01908-f004:**
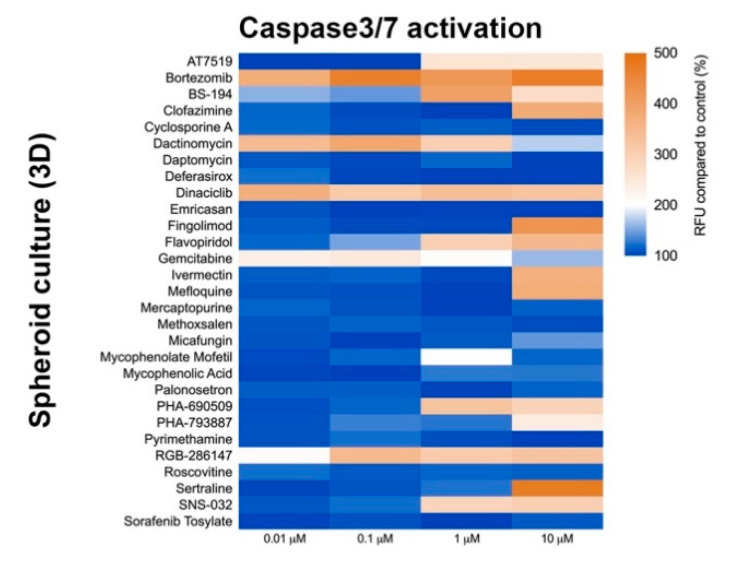
Caspase-3/7 activation assay after compound exposure. Heat map of relative fluorescence increase after 48 h compound exposure. CellEvent^TM^ reagent was used to measure Caspase-3/7 activation. Graph shows fluorescence increase relative to control (DMSO-treated).

**Figure 5 ijms-22-01908-f005:**
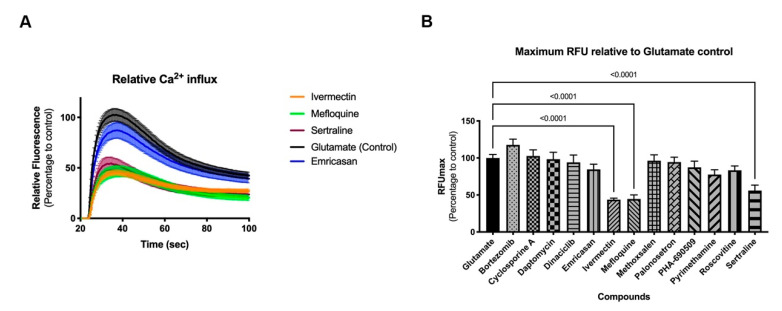
Functional response of 2D neural cultures to glutamate stimulus. Graphs show relative fluorescence response over time (**A**) and maximum relative fluorescence (**B**) of glutamate induced activity. Eight-week-old 2D neural cultures were exposed with selected compounds for 30 min before stimulus with glutamate and functional response was recorded using FLIPR^TETRA^. Standard deviation is shown in both graphs. Number of replicates per compound = 14.

**Figure 6 ijms-22-01908-f006:**
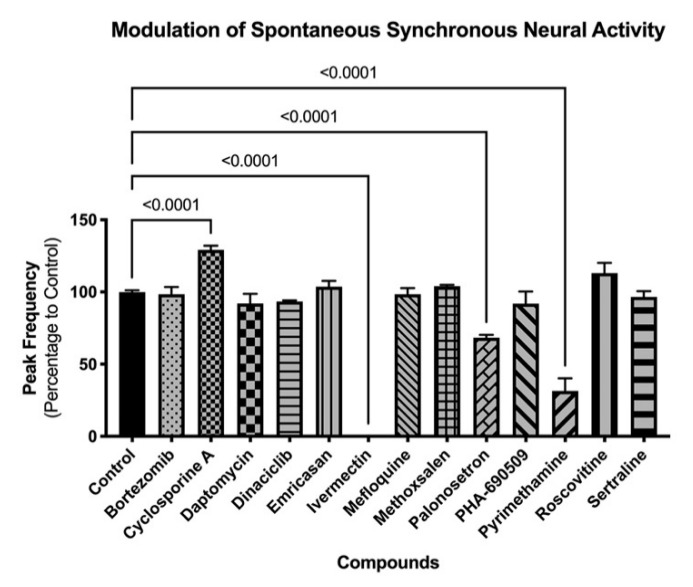
Modulation of spontaneous neural activity in 3D cultures. Graph shows modulation of spontaneous synchronized activity (peak frequency) after compound exposure. Spheroids were exposed to compounds for 30 min, and functional response was recorded using FLIPR^TETRA^. Standard deviation is shown. Number of replicates per compound = 4.

**Table 1 ijms-22-01908-t001:** Selected Compounds and References.

Compound	Clinical or Intended Use	MOA	Group	Reference
AT7519	Cancer treatment	CDK Inhibitor	Investigational	[12]
Bortezomib	Cancer treatment	Proteosome Inhibitor	Approved	[13]
BS-194	Cancer treatment	CDK Inhibitor	Investigational	[12]
Clofazimine	Leprosy treatment	Precise Mechanism Unknown	Approved	[13]
Cyclosporine A	Prophylaxis of organ transplant rejection	Calcineurin Inhibitor	Approved	[13]
Dactinomycin	Cancer treatment	DNA Topoisomerase Inhibitor	Approved	[13]
Daptomycin	Treatment of skin infections	Destabilization of Bacterial Cell Membrane	Approved	[13]
Deferasirox	Treatment of chronic iron overload	Iron Chelator	Approved	[13]
Dinaciclib	Cancer treatment	CDK Inhibitor	Investigational	[12]
Emricasan	Treatment of Hepatitis, liver disease, and organ transplantation	Caspase Inhibitor	Investigational	[12]
Fingolimod	Multiple sclerosis	Sphingosine-1-phosphate Receptor Modulator	Approved	[13]
Flavopiridol	Cancer treatment	CDK Inhibitor	Investigational	[12]
Gemcitabine	Cancer treatment	Thymidylate Synthetase Inhibitor	Approved	[13]
Ivermectin	Broad-spectrum anti-parasite medication	Agonist of Chloride Ion Channels in Invertebrates	Approved	[13]
Mefloquine	Treatment of Malaria	Precise Mechanism Unknown	Approved	[13]
Mercaptopurine	Cancer treatment	Hypoxanthine-guanine Phosphoribosyltransferase Inhibitor	Approved	[13]
Methoxsalen	Treatment of psoriasis and vitiligo	DNA Intercalation	Approved	[13]
Micafungin	Treatment of fungal infections	Inhibits the synthesis of beta-1,3-D-glucan	Approved	[13]
Mycophenolate Mofetil	Prophylaxis of organ transplant rejection	Inosine 5’-Monophosphate Dehydrogenase Inhibitor	Approved	[13]
Mycophenolic Acid	Prophylaxis of organ transplant rejection	Inosine 5’-Monophosphate Dehydrogenase Inhibitor	Approved	[13]
Palonosetron	Prevention of acute nausea and vomiting from chemotherapy	5-Hydroxytryptamine Receptor Antagonist	Approved	[13]
PHA-690509	Cancer treatment	CDK Inhibitor	Investigational	[12]
PHA-793887	Cancer treatment	CDK Inhibitor	Investigational	[12]
Pyrimethamine	Treatment of Malaria and Toxoplasmosis	Dihydrofolate Reductase Inhibitor	Approved	[13]
RGB-286147	Cancer treatment	CDK Inhibitor	Investigational	[12]
Roscovitine	Cancer treatment	CDK Inhibitor	Investigational	[12]
Sertraline	Antidepressant	Reuptake of Serotonin Inhibitor	Approved	[13]
SNS-032	Cancer treatment	CDK Inhibitor	Investigational	[12]
Sorafenib Tosylate	Cancer treatment	Inhibits Several Intracellular and Cell Surface Kinases	Approved	[13]

**Table 2 ijms-22-01908-t002:** Compounds selected for functional investigation and their no observed adverse effect level (NOAEL) concentrations.

Compound	NOAEL Concentration (μM)
***Bortezomib***	0.001
***Cyclosporine A***	1
***Daptomycin***	10
***Dinaciclib***	0.001
***Emricasan***	10
***Ivermectin***	1
***Mefloquine***	1
***Methoxsalen***	10
***Palonosetron***	10
***PHA-690509***	0.1
***Pyrimethamine***	10
***Roscovitine***	10
***Sertraline***	1

**Table 3 ijms-22-01908-t003:** List of primary antibodies used in this study.

Antibody	Brand	Concentration
Nestin	Abcam	1:250
Sox1	R&D Systems	1:400
MAP2	Millipore	1:200
GFAP	Abcam	1:500
Synapsin I	Abcam	1:500
NeuN	Millipore	1:200
Tuj 1	Millipore	1:500

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
