# Peer review of "Human iPSC-Derived 2D and 3D Platforms for Rapidly Assessing Developmental, Functional, and Terminal Toxicities in Neural Cells"

_ijms, 2021, doi:10.3390/ijms22041908_

Round 1

Reviewer 1 Report

Summary:  In this manuscript the authors present screening platforms based on human iPSCs to evaluate dose-dependent neurotoxicity profiles of therapeutics.  The manuscript is very clearly written, and laboratory method well detailed, and the experimental platform is well motivated.  Lack of detail regarding statistical analysis is disappointing, as are some of the inferential claims made without any formal analysis.

Major Comments:

The statistical analysis section is very underdeveloped.  It would be at minimum useful to describe the statistical analyses used for the comparisons illustrated in Figures 5B and 6.  Presumably an ANOVA with Dunnett correction for multiple testing?

Many inferences are made without really any formal analytical methodology, but rather appear to be by passing visual “sniff tests” via heatmap comparisons.  Statements like “Culture format does not influence cellular toxicity response” is fairly strong without any evaluation beyond visualization of results.  It seems like a lot could be addressed with correlation/concordance analysis (evaluating comparability) or ANOVA-type analyses (evaluating differences).  It seems like the authors expect many findings to be concluded as merely “obvious”.  However, it seems like all the necessary data are there, so as long as the underlying sample size is robust, it should be fairly easy to run relevant analyses.

Care should also be made with the term “significant” – this generally accompanies many of the  inferential statements but sems to be used in context of degree rather than in a statistical context.  The term should only be reserved with respect to statistical inference to avoid confusion.

Author Response

The statistical analysis section is very underdeveloped.  It would be at minimum useful to describe the statistical analyses used for the comparisons illustrated in Figures 5B and 6.  Presumably an ANOVA with Dunnett correction for multiple testing?

We thank the reviewer for all comments. For both 2D (with Kainic Acid and Glutamate) and 3D (spontaneous) analyses, we performed one-Way ANOVA (and Nonparametric or Mixed). T-tests were performed to compare similar aged cultures across 2D or 3D formats. We have now uploaded the spreadsheet (Supplemental Data 1) containing the statistical data and apologize for omitting it previously.

Many inferences are made without really any formal analytical methodology, but rather appear to be by passing visual “sniff tests” via heatmap comparisons.  Statements like “Culture format does not influence cellular toxicity response” is fairly strong without any evaluation beyond visualization of results.  It seems like a lot could be addressed with correlation/concordance analysis (evaluating comparability) or ANOVA-type analyses (evaluating differences).  It seems like the authors expect many findings to be concluded as merely “obvious”.  However, it seems like all the necessary data are there, so as long as the underlying sample size is robust, it should be fairly easy to run relevant analyses.

We appreciate the reviewer noticing the lack of statistical detail surrounding the heatmaps. The visual cues provided by the heat map assessments have been bolstered with a quantitative description for Supplemental Figure 2A (text lines 121 – 128) and t-test analyses for Figures 2 and 3 (text lines 144 – 148, 165 - 168),

Care should also be made with the term “significant” – this generally accompanies many of the inferential statements but seems to be used in context of degree rather than in a statistical context.  The term should only be reserved with respect to statistical inference to avoid confusion.

We thank the reviewer for this observation. We have changed the wording or parenthetically added in the appropriate p values and associated statistical test noting significance. All changes to the text, including changes to the statistical information in the Methods section and body of the text, are noted in red.  

Reviewer 2 Report

In this work the authors describe a new methods for testing potentially neurotoxic compounds in relation with developmental stages of target cells, using hiPSC. Moreover the study has been performed both in 2D and 3D cultures.

This study is very well and clearly presented. I believe that the impact of such new method is very high, since it render possible to test neurotossicity in a human developing neural model, completing the panel of toxicity tests already in use.

My only concern is that this test may be too sensitive, eliminating very effective drugs that in vivo may not have any adverse effect on CNS, because of, for istance, the presence of the hemato-encephalic barrier.

I would ask to add a specific comment on this aspect, that, I believe, may be very important.

Author Response

My only concern is that this test may be too sensitive, eliminating very effective drugs that in vivo may not have any adverse effect on CNS, because of, for istance, the presence of the hemato-encephalic barrier.

I would ask to add a specific comment on this aspect, that, I believe, may be very important.

We thank the reviewer for this comment regarding the sensitivity of our system, especially as  it relates to the lack of a blood-brain barrier (BBB) in our test system. While the BBB is an important factor in understanding how much compound is exposed to neural tissue, it does not play a direct role in determining how neural tissue reacts to a specific concentration nor does it impact the conclusion that progenitor cells are more susceptible than their more mature counterparts. We have acknowledged the topic in lines 310 – 320.

All changes to the text are noted in red.  

Round 2

Reviewer 1 Report

The authors have largely addressed my concerns.  The stats methods section could still be fleshed out a bit (since some of the analysis methods are embedded in the results instead).

Author Response

Once again, we thank the reviewer for the helpful comments. We have revised the statistical analysis section of the Methods to include information regarding the circumstances under which each statistical test was performed (please see the attachment). To make the revisions easier to find, we have colored these updates blue. 

Gratefully,

Mary Anne Lofstrom on behalf of all of the authors